# Motor Performance in Association with Perceived Loneliness and Social Competence in 11-Year-Old Children Born Very Preterm

**DOI:** 10.3390/children9050660

**Published:** 2022-05-04

**Authors:** Minttu Helin, Max Karukivi, Leena Haataja, Päivi Rautava, Niina Junttila, Susanna Salomäki, Liisa Lehtonen, Sirkku Setänen

**Affiliations:** 1Department of Pediatric Neurology, Turku University Hospital, 20521 Turku, Finland; sirkku.setanen@utu.fi; 2Department of Pediatric Neurology, University of Turku, 20521 Turku, Finland; 3Department of Adolescent Psychiatry, Turku University Hospital, 20700 Turku, Finland; max.karukivi@utu.fi; 4Department of Adolescent Psychiatry, University of Turku, 20700 Turku, Finland; 5Department of Pediatric Neurology, University of Helsinki, 00014 Helsinki, Finland; leena.haataja@hus.fi; 6Pediatric Research Center, Helsinki University Hospital, 00290 Helsinki, Finland; 7Turku Clinical Research Centre, Turku University Hospital, 20521 Turku, Finland; paivi.rautava@tyks.fi; 8Public Health, University of Turku, 20014 Turku, Finland; 9Department of Teacher Education, University of Turku, 20014 Turku, Finland; niina.junttila@utu.fi; 10Department of Psychology, University of Turku, 20014 Turku, Finland; sumasa@utu.fi; 11Department of Pediatrics, Turku University Hospital, 20521 Turku, Finland; liisa.lehtonen@utu.fi; 12Department of Pediatrics, University of Turku, 20521 Turku, Finland

**Keywords:** CP, DCD, motor impairment, long-term follow-up, Movement ABC-2, PNDL, MASCS, children born preterm, regional cohort study

## Abstract

Background: Very preterm birth may affect motor performance and social competence up to adulthood. Our objective was to describe perceived loneliness and social competence in children born very preterm in relation to motor impairment. Methods: 165 children born very preterm (birth weight ≤ 1500 g and/or gestational age < 32 weeks) were assessed at 11 years of age. Cerebral palsy (CP) was diagnosed by 2 years of age. At 11 years of age, motor outcome was assessed using the Movement Assessment Battery for Children—Second edition (Movement ABC-2). Loneliness was evaluated by using the Peer Network and Dyadic Loneliness scale and social competence by using the Multisource Assessment of Children’s Social Competence Scale. Results: In total, 6 (4%) children had CP, 18 (11%) had Developmental Coordination Disorder (DCD) (Movement ABC-2 ≤ 5th percentiles), and 141 (85%) had typical motor development. There was no correlation between percentiles for total scores of the Movement ABC-2 and perceived loneliness or social competence when the children with motor impairment (CP or DCD) were excluded. Children with DCD reported less perceived loneliness, but more problems with social competence compared to children with CP. Conclusions: It is important to recognize children born very preterm with DCD to provide interventions and support services to prevent social exclusion.

## 1. Introduction

Children born very preterm have an increased risk of long-lasting health and developmental problems [1]. They are at a higher risk for motor impairment than children born full term [2,3,4,5]. Preterm birth is the most significant risk factor for cerebral palsy (CP) [6,7]. The prevalence of CP ranges from 7% to 20% in children born very preterm [8,9,10]. Children with CP have been shown to have less social activity with their peers, especially when their independent movement is limited [11]. Young adults at 20–22 years of age with CP face challenges maintaining social networks and independency, because of limitations in mobility, communication, and their dependency on family members [12].

Developmental coordination disorder (DCD) is more common than CP in children born very preterm [13,14]. These motor problems seem to continue into adulthood [15,16]. DCD is defined as a motor impairment interfering with academic achievement or the activities of daily living, which cannot be explained by any obvious neurological and structural abnormality or intellectual disability [17]. DCD is typically associated with poor balance, coordination, and handwriting. The prevalence of DCD varies from 5% to 6% in school-aged children and from 8% to 51% in children born very preterm [2,14,17,18,19]. Children with DCD, in a mixed gestational age population, are known to participate less often in sports and regular physical activities compared to their peers with typical motor development, leading to difficulties with peer relationships and the risk of social exclusion [20]. 

Children born very preterm have an increased risk for social functioning problems, including low social competence, autism spectrum symptoms, social withdrawal, internalizing behavior, and attention deficits [1,21,22,23,24]. It has been stated that children born extremely preterm have fewer friends and they spend less time with friends at 12 years of age compared to children born full term [25]. The peer relationship difficulties of children born extremely preterm have been reported to continue from childhood to adulthood [26]. Social competence consists of emotional, cognitive, and behavioral skills needed for successful social adaptation. Social loneliness has been defined as an emotional response resulting from the perceived absence of a social network, or the feeling that one is not part of a group. Emotional loneliness has been defined as an unpleasant emotional response to the lack, or perceived lack, of an intimate dyadic friendship, apart from the absence of social networks [27,28]. Loneliness poses significant mental and physical health problems in adolescence and adulthood [29,30,31]. Social, emotional, cognitive, and behavioral skills are needed for successful social adaptation. Social loneliness has been defined as an emotional response resulting from the perceived absence of a social network, or the feeling that one is not part of a group. Emotional loneliness has been defined as an unpleasant emotional response to the lack, or perceived lack, of an intimate dyadic friendship, apart from the absence of social networks [27,28]. Loneliness poses significant mental and physical health problems in adolescence and adulthood [29,30,31]. 

Developmental outcomes of children born very preterm differ from their peers born full term [32,33,34,35]. Earlier studies on association between motor performance and loneliness or social competence have focused on children born full term or mixed gestational age populations. We have recently reported that very preterm birth may affect social competence at 11 years of age [35]. It is not known if motor performance is associated with loneliness in children born very preterm. This knowledge would have clinical implications and the need to provide support services and interventions to prevent social exclusion. 

Our aim was to study the association between motor performance and perceived loneliness and social competence as a combination of co-operative skills, empathy, impulsivity, and disruptiveness in 11-year-old children born very preterm. We hypothesized that motor impairment is associated with perceived loneliness and social competence. 

## 2. Materials and Methods

### 2.1. Participants

This study is part of the Finnish prospective multidisciplinary PIPARI Study (The Development and Functioning of very low weight infants from Infancy to School Age). The participants were born to Finnish- or Swedish-speaking families in the Turku University Hospital between 2001 and 2006. The inclusion criteria were birth weight ≤ 1500 g and gestational age < 37 weeks. From the beginning of 2004, the inclusion criteria were expanded to include all infants born < 32 gestational weeks, despite the birth weight. In addition, the participants in the present study had to have the following evaluations at 11 years of age (1) Movement Assessment Battery for Children—Second Edition (Movement ABC-2) [36] for motor outcome, (2) Peer Network and Dyadic Loneliness scale (PNDL) [28,37] for perceived loneliness, and (3) Multisource Assessment of Children’s Social Competence Scale (MASCS) [38,39] for perceived social competence. Information about the children’s clinical diagnosis were collected from families during the study visit and from medical records. The flowchart of the participants is shown in Figure 1.

### 2.2. Motor Outcome 

The motor outcome was categorized as typical motor development (Movement ABC-2 > 5th percentiles) or motor impairment (DCD or CP). The diagnosis of CP was made by a senior child neurologist by 2 years of corrected age based on the classification proposed by Himmelmann et al. [40]. The Gross Motor Function Classification System (GMFCS) was used to grade the functional severity [41]. At 11 years of age, the motor outcome was assessed using the Movement ABC-2 for age band 3 (11–16 years of age). The neurological and motor assessments were performed by three physicians trained to use the assessment methods. The Movement ABC-2 includes three subscales: aiming and catching (three items), manual dexterity (three items), and balance (three items). The test was scored using raw scores, which were converted to total standard scores and percentile scores according to the test manual [36]. The norms for 11-year-old children were used [36]. Based on the manual, a total test score ≤ 56 (≤5th percentile) indicated DCD, whereas higher scores indicated typical motor development. A strict cut-off of the 5th percentile was used to define clinically significant motor impairment other than CP [13]. Other neurological conditions affecting motor development were excluded by the Touwen neurological examination [42], which was performed during the same visit. Children with full-scale intelligence quotient (IQ) < 70 (*n* = 6) were able to follow the given instructions and completed the Movement ABC-2 appropriately. All the Movement ABC-2 assessments and Touwen neurological examinations were video-recorded and saved in case of the need for reassessment. 

### 2.3. Loneliness at 11 Years of Age

Loneliness was evaluated by using the PNDL [28,37]. This scale measures social loneliness as a lack of perceived peer networks and emotional loneliness as a subjective feeling of absence of close friendship. The first part contains five questions about social loneliness, and the second part includes five questions about emotional loneliness. In every question, there are two propositions, and the children select the one that best describes their situation, such as, “Some kids feel like they really fit in with other kids but some kids don’t feel like they fit in with other kids”. After that, the children evaluate whether the proposition fit them “quite well” or “very well”. Every item is scored on a scale from one to four. After reversing the negative items, the higher scores indicate higher loneliness. Both subscales of loneliness were analyzed as a continuous variable. The Cronbach’s alpha for PNDL was 0.64 for social loneliness and 0.66 for emotional loneliness [28].

### 2.4. Social Competence at 11 Years of Age

Social competence was evaluated by using the MASCS [39]. It is an instrument for assessing social competence of elementary school children, considering the multiple perspectives of self, peers, teachers, and parents. Its development is based on the School Social Behavior Scale, which is meant to be used by teachers. The questionnaire has been validated for self, peer, teacher, and parent ratings in a sample of Finnish children [38,39]. MASCS consist of 15 items evaluating cooperative skills (5 items), empathy (3 items), impulsivity (3 items), and disruptiveness (4 items). Every item contains a four-point summative scale. In this study, only the self-assessment-part of the MASCS was used. The Finnish MASCS manual was used to score the questionnaire [38]. Higher scores indicate higher co-operative skills, empathy, impulsivity, or disruptiveness. All the subscales of social competence were analyzed as continuous variables. The Cronbach’s alpha for MASCS was 0.65 for co-operation skills, 0.73 for empathy, 0.67 for impulsivity, and 0.60 for disruptiveness [39].

### 2.5. Statistical Analysis

The normality of the distributions was assessed both graphically and with the Shapiro–Wilk test (Table 1). The PNDL and MASCS subscales were described by means, standard deviations and minimum and maximum values. Continuous variables were compared between study children and drop-outs using the independent sample *t*-test or Mann–Whitney *U*-test and comparisons between two categorical variables were done using the Pearson Chi Square or Fisher’s exact test, as appropriate. Pearson’s correlation was used to study the univariate associations between two continuous variables (percentiles for total scores of the Movement ABC-2 and PNDL subscales, Movement ABC-2, and MASCS subscales). Motor performance was analyzed both as a continuous variable (percentiles for total scores of the Movement ABC-2) and as categorized variables typical motor development (Movement ABC-2 > 5th percentiles), DCD (Movement ABC-2 ≤ 5th percentiles), or CP (clinical diagnosis). The differences in the mean values between the groups (CP, DCD, and typical motor development) were analyzed using univariate analysis of variance (ANOVA). Statistical analyses were performed using SPSS version 28. *p*-value < 0.05 was considered as statistically significant.

## 3. Results

A total of 165 of 219 surviving study participants were assessed at 11 years of age (Figure 1). There were 143 children born <32 weeks of gestation and 154 children born ≤1500 g. The neonatal characteristics were compared between the final study group and drop-outs. The children participating at 11 years of age were more often born from multiple pregnancies than those who had dropped out (Table 1). In total, 6 (3.6%) children developed CP and 18 (10.9%) children developed DCD (Table 2). Only 1 child out of the 18 with DCD had a clinical diagnosis according to the patient records. A total of 14 (8.4%) children had a full-scale IQ < 70, and 3 (1.8%) children had severe hearing impairment. There were no children with severe visual impairment. In total, 9 of the 15 (60%) children with CP had dropped out by 11 years of age.

The PNDL scores were available for perceived social loneliness for 159 (96%) children and for perceived emotional loneliness for 157 (95%) children. The mean (SD) score for social loneliness was 7.9 (2.4) and for emotional loneliness 7.7 (2.5). The percentiles for the total scores of the Movement ABC-2 correlated negatively with perceived social loneliness (r = −0.2, *p* = 0.049), when all the study children were included. The correlation did not remain statistically significant when the groups (CP, DCD, typical motor development) were analyzed separately (Table 3). There was no statistically significant correlation between the percentiles for the total scores of the Movement ABC-2 and perceived emotional loneliness. 

MASCS scores were available for co-operative skills for 159 (96%) children, for empathy for 164 (99%) children, for impulsivity for 163 (99%) children, and for disruptiveness for 162 (98%) children. The mean (SD) score for co-operative skills was 10.1 (1.3), for empathy 6.0 (0.8), for impulsivity 4.1 (1.1), and for disruptiveness 3.7 (0.9). There was no statistically significant correlation between the percentiles for the total scores of the Movement ABC-2 and perceived social competence (Table 3). In post-hoc analysis, boys born very preterm were more likely to report less empathy (boys mean 5.7 vs. girls mean 6.3, *p* < 0.001) and more disruptiveness (boys mean 3.9 vs girls mean 3.5, *p* = 0.002) compared to girls born very preterm. 

Table 4 shows the mean (SD) and *p*-values of the PNDL and MASCS subscales for children with DCD, CP, and typical motor development. Children with CP had higher scores for perceived social and emotional loneliness compared to children with DCD and with typical motor development. Regarding social competence, children with CP had better scores in all subscales (co-operation, empathy, impulsivity, and disruptiveness) compared to children with DCD or with typical motor development.

## 4. Discussion

This follow-up study of a regional cohort of Finnish children born very preterm showed that motor performance did not correlate with perceived loneliness or social competence in children with typical motor development at 11 years of age. Children with DCD reported less perceived loneliness, but more problems with social competence compared to children with CP or with typical motor development. The absolute differences in the PNDL and MASCS results between the groups were minor. Whether these differences have clinical importance is not definite.

All the children with CP were diagnosed by 2 years of age and the diagnoses were confirmed at 11 years of age [44]. In Finland, all children with CP participate in regular clinical follow-ups and have individually tailored rehabilitation (physiotherapy, occupational therapy, and/or speech therapy). The routine clinical follow-ups after very preterm birth ends at the age of 2 years of corrected age. Therefore, children with motor impairment, such as DCD, are presumed to have had primary diagnostic assessments and support from the school health care services. Evidently, DCD is not recognized by clinicians working in primary care services. This might explain our finding that children with DCD experienced more problems in social competence compared to children with CP. The number of children with CP and DCD was small, which may limit the generalizability of our results. Future research is needed regarding motor performance and perceived loneliness and social competence in a larger population of children born very preterm. 

Children with DCD, in a mixed gestational age population, have been shown to suffer from social isolation, problems with social skills, and difficulties in peer relationships [20,45], which is in line with our findings in this very preterm population. It has also been shown that children with DCD, in a mixed gestational age population, have anxiety and depression more often than their peers with typical motor development [46]. We have previously shown that children born very preterm with DCD had a lower perceived quality of life and more often cognitive impairment (full-scale IQ < 70) compared with children born very preterm with typical motor development at 11 years of age [14]. We were not able perform any statistical analyses with adjustments because of the limited number of children with a motor impairment. In future studies, it would be important to consider the cognitive outcome as a covariate when analyzing the effect of motor performance on perceived loneliness and social competence in children born very preterm.

In the present study, boys perceived that they had more problems in social competence than girls. There was no sex difference in perceived social or emotional loneliness. In our recent study, most children born very preterm reported a high or average social functioning profile irrespective of sex [35]. Boys born very preterm reported low social functioning less often than same-sex controls born full term. In contrast, girls born very preterm reported high social functioning less often than same-sex controls. Sex profile differences were found to be smaller in children born very preterm than in the controls [35]. Further research is needed to understand the effects of very preterm birth on the development of social competence in both genders.

The strength of our study was its relatively high follow-up rate of 71% up to 11 years of age. The Touwen neurological examination [42] was used to support the definition of DCD and the motor examinations were performed with the most recent version of Movement ABC-2. A possible limitation was that motor assessments could not be done according the European Academy of Childhood Disability’s new guidelines, as these were published afterwards [13]. However, these new guidelines were not available at the time when the data were collected. Although the number of whole study children was competent, the number of children who developed motor impairment (CP or DCD) was small and many of them did not participate at the age of 11 years, limiting the possibility to use the statistical analyses in describing the differences between the groups. All the children with DCD did not complete both the PNDL and MASCS assessments. There are no norms or cut-offs for children born very preterm regarding the results of the PNDL and the MASCS. One limitation in our study was that social competence was evaluated by self-assessments; evaluation by parents, teachers, and peers may have provided a broader view. However, children’s self-assessments have been shown to correlate with assessments by parents, teachers, and peers [39]. The Crohnbach’s alphas were moderate for both subscales of PNDL as well as the co-operation skills, impulsivity, and disruptiveness measured by MASCS. Finally, we had no term-born controls. 

The current study expands our knowledge of the relation between motor impairment and perceived loneliness and social competence at 11 years of age of children born very preterm. Adolescence is a transitional phase from childhood to adulthood. Understanding this is crucial to securing the optimal development of children born very preterm. The feelings of social isolation and loneliness may lead to a limited number of activities and even social exclusion. In long-term follow-ups of very preterm born children, it would be important to also recognize other motor impairments than CP; this would ensure that timely, individually tailored support services and interventions were provided in order to prevent social exclusion. 

## Figures and Tables

**Figure 1 children-09-00660-f001:**
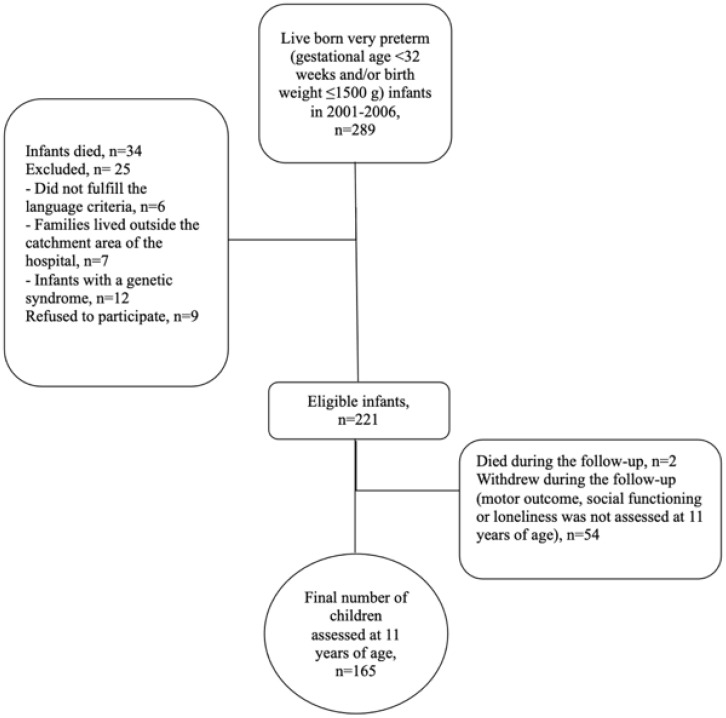
Flowchart of the participants.

**Table 1 children-09-00660-t001:** Background characteristics of the study children born very preterm (birth weight ≤1500 g and or gestational age < 32 weeks) compared with the characteristics of the drop-outs (including two children who died). Continuous variables were compared between the study children and the drop-outs using the independent sample *t*-test or Mann–Whitney *U*-test and comparisons between two categorical variables were done using the Pearson Chi Square or Fisher’s exact test, as appropriate.

	Study Children, *n* = 165	Drop-Outs, *n* = 56	*p*-Value
Gestational age, mean (SD) [minimum, maximum], week	29.0 (2.7) [23 + 0, 35 + 6]	29.0 (2.8) [23 + 5, 35 + 1]	1.0
Birth weight, mean (SD) [minimum, maximum], grams	1128.7 (315.9) [400.0, 2120.0]	1181.3 (369.5) [565.0, 1970.0]	0.3
Birth weight z-score, mean (SD) [minimum, maximum]	−1.4 (1.5) [−4.9, 3.4]	−1.3 (1.3) [−3.9, 2.0]	0.6
Small for gestational age (<−2 SD), *n* (%)	54 (32.7)	14 (25.0)	0.3
Male, *n* (%)	92 (55.8)	33 (58.9)	0.7
Cesarean delivery, *n* (%)	97 (58.8)	37 (66.1)	0.3
Multiple birth, *n* (%)	58 (35.2)	9 (16.1)	0.007
Bronchopulmonary dysplasia, *n* (%)	20 (12.1)	8 (14.3)	0.7
Operated necrotizing enterocolitis, *n* (%)	6/162 (3.7)	4/55 (7.3)	0.3
Sepsis, *n* (%)	30 (18.2)	7 (12.5)	0.3
Laser-treated retinopathy of prematurity, *n* (%)	4/154 (2.6)	4/53 (7.5)	0.2
Major brain pathologies in magnetic resonance imaging at term age * *n* (%)	39/160 (24.4)	17/55 (30.9)	0.3
Mother’s education > 12 years, *n* (%)	67/155 (43.2)	18/50 (36.0)	0.4
Father’s education > 12 years, *n* (%)	39/154 (25.3)	9/48 (18.8)	0.4

* Setänen et al. have published in 2013 the specific MRI protocol and details about the classification of the findings [43].

**Table 2 children-09-00660-t002:** Characteristics of children born very preterm with cerebral palsy (CP) and developmental coordination disorder (DCD).

	Children with CP, *n* = 6	Children with DCD, *n* = 18
Gross Motor Function Classification System I, *n* (%) II, *n* (%) III, *n* (%)	2 (33) 3 (50) 1 (17)	
Boys, *n* (%)	4 (67)	17 (94)
Full-scale intelligence quotient < 70, *n* (%)	2 (33)	7 (39)
Severe hearing impairment, *n* (%)	0 (0)	1 (6)

**Table 3 children-09-00660-t003:** Outcome characteristics of the 11-year-old children born very preterm. Motor outcome was assessed with the Movement Assessment Battery for Children, Second Edition (Movement ABC-2). Social functioning was assessed with the Multisource Assessment of Children’s Social Competence Scale (MASCS) and loneliness was assessed with the Peer Network and Dyadic Loneliness Scale (PNDLS). Pearson’s correlation was used to study the univariate associations between two continuous variables. *n* = number of assessed children, r = Pearson correlation coefficient.

Outcome Variable	Movement ABC-2 Children with CP *n* = 6	Movement ABC-2 Children with DCD, *n* = 18	Movement ABC-2 Children Typical Motor Development, *n* = 141
PNDLS			
Social loneliness	r = −0.6, *p* = 0.2	r = 0.3, *p* = 0.3 ^c^	r = −0.1, *p* = 0.4 ^c^
Emotional loneliness	r = 0.4, *p* = 0.5 ^a^	r = −0.3, *p* = 0.2 ^a^	r = 0.0, *p* = 0.6 ^e^
MASCS			
Co-operation skills	r = 0.2, *p* = 0.7	r = −0.2, *p* = 0.4 ^b^	r = 0.0, *p* = 0.7 ^d^
Empathy	r = −0.3, *p* = 0.5	r = −0.3, *p* = 0.3	r = 0.1, *p* = 0.5 ^a^
Impulsivity	r = 0.5, *p* = 0.3	r = 0.06, *p* = 0.8	r = 0.0, *p* = 0.6 ^b^
Disruptiveness	r = 0.4, *p* = 0.4	r = −0.2, *p* = 0.4	r = −0.1, *p* = 0.4 ^c^

^a^ data missing for one child. ^b^ data missing for two children. ^c^ data missing for three children. ^d^ data missing for four children. ^e^ data missing for six children.

**Table 4 children-09-00660-t004:** Perceived loneliness assessed with the Peer Network and Dyadic Loneliness scale (PNDL) and social competence assessed with the Multisource Assessment of Children’s Social Competence Scale (MASCS) in children born very preterm with cerebral palsy (CP), with developmental coordination disorder (DCD) and with typical motor development at 11 years of age. Higher scores indicate higher loneliness, co-operative skills, empathy, impulsivity, or disruptiveness. ANOVA was used to study the differences in the mean values between the groups.

Outcome Variable	Children with CP, *n* = 6 Mean (SD)	Children with DCD, *n* = 18 Mean (SD)	Children with Typical Motor Development, *n* = 141 Mean (SD)	*p*-Value
PNDL				
Social loneliness	9.0 (1.5)	8.9 (2.7) ^c^	7.8 (2.3) ^c^	0.1
Emotional loneliness	8.8 (1.9) ^a^	8.6 (3.0) ^a^	7.6 (2.4) ^e^	0.2
MASCS				
Cooperation skills	10.9 (1.6)	10.3 (1.5) ^b^	10.1 (1.3) ^d^	0.3
Empathy	6.4 (0.8)	5.8 (0.9)	6.0 (0.8) ^a^	0.3
Impulsivity	2.9 (1.0)	4.5 (1.0)	4.1 (1.1)^b^	0.008 *
Disruptiveness	2.9 (0.4)	3.9 (0.2)	3.7 (0.1)^c^	0.1

* Tukey-corrected *p*-values; *p* = 0.005 between the children with CP and with DCD, *p* = 0.02 between the children with CP and with typical motor development, *p* = 0.3 between the children with DCD and with typical motor development. ^a^ data missing for one child. ^b^ data missing for two children. ^c^ data missing for three children. ^d^ data missing for four children. ^e^ data missing for six children.

## Data Availability

The data that support the findings of this study are available on request from the corresponding author. The data are not publicly available.

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
