# Peer review of "Motor Performance in Association with Perceived Loneliness and Social Competence in 11-Year-Old Children Born Very Preterm"

_children, 2022, doi:10.3390/children9050660_

Round 1
Reviewer 1 Report
Longitudinal studies of preterm infants are of great scientific and practical interest. In this article, the authors present a study on association of motor performance with social competence and perceived loneliness in 11-years-old children born very preterm. The concept and the idea of the study are very relevant, but unfortunately, they are still poorly presented in the article. Also, before presenting comments and recommendations for the authors, it should be noted that the article has many technical shortcomings that make it difficult to read: non-uniform font, "clickable" links to terms, tables inserted into the text as non-editable pictures.
The Introduction is very comprehensive but short. It is desirable to elaborate on the results of previous studies in order to emphasize the novelty and originality of the presented research.
In our opinion, there is a contradiction between the title of the article, the purpose, the hypothesis and, accordingly, the results presented below.
- Title is “Motor impairment in association with perceived loneliness and social competence in 11-year-old children born very preterm”
- The purpose “was to study the association between motor performance and perceived loneliness and social competence as a combination of co-operative skills, empathy, impulsivity and disruptiveness in 11-year-old children born very preterm”.
- The hypothesis was “that motor impairment is associated with perceived loneliness and social competence”.
This raises the question of what the authors are studying: motor impairment or motor performance?
The Material and Methods section provides a detailed description of a sample of 165 children, as well as a description of 3 diagnostic tools (Motor outcome, Loneliness, Social competence) and statistical analysis. In our opinion, this is the most balanced section in the article.
The Results section includes tables on:
- Background characteristics comparison between the study children (n = 165) and the drop-outs (n = 56)
- Characteristics of children born very preterm with cerebral palsy (n = 6) and developmental coordination disorder (n = 18)
- Perceived loneliness and social competence in children born very preterm with cerebral palsy (n = 6) and with developmental coordination disorder (n = 18)
Thus, the results are presented comparing perceived loneliness and social competence only in children with CP and DCD; such results for the rest of the participants are not presented. Moreover, this section does not include the results of the Pearson correlation analysis, which is listed in the subsection 2.5. In our opinion, it is need the correlation analysis between the studied variables (n = 165) that is necessary to achieve the purpose of the research: “to study the association between motor performance and perceived loneliness and social competence in 11-year-old children born very preterm”.
Accordingly, it is not clear on what basis the authors write in the Discussion that “in contrast to our hypothesis, motor performance did not correlate with perceived loneliness or social competence in children with typical motor development at 11 years of age.” Unfortunately, the article does not contain the results of the correlation analysis, as well as the results of the comparison between boys and girls, which are discussed below.
In conclusion, I can say that the authors have good empirical data, but this version of the article does not provide a clear analysis of these data and a description of the results in accordance with the purpose and hypothesis of the study.
Author Response
Dear Reviewer,
 
Thank you for improving the manuscript with your thorough comments. We have now answered the comments below. Accordingly, we have revised the manuscript carefully. 
 
 
Reviewers Comments to Author 
 
 
Point 1: The concept and the idea of the study are very relevant, but unfortunately, they are still poorly presented in the article. Also, before presenting comments and recommendations for the authors, it should be noted that the article has many technical shortcomings that make it difficult to read: non-uniform font, "clickable" links to terms, tables inserted into the text as non-editable pictures. 
 
Answer 1: We apologize all the technical shortcomings that were related to compatible problems with apple computer and the template. All pictures and tables are available also as separate formats. Template advised to insert pictures into the text. We have revised the manuscript with pc-computer so there should be no more technical problems.  
 
 
Point 2: The Introduction is very comprehensive but short. It is desirable to elaborate on the results of previous studies in order to emphasize the novelty and originality of the presented research.
 
Answer 2: We have added more detailed information of previous studies. All additions to the text are marked in red:  
It has been stated that children born extremely preterm have less friends and they spend less time with friends at 12 years of age compared to children born full-term [25]. The peer relationship difficulties of children born extremely preterm have been reported to continue from childhood to adulthood [26].
 
Point 3: In our opinion, there is a contradiction between the title of the article, the purpose, the hypothesis and, accordingly, the results presented below.
Answer 3: We appreciate this observation. We have unified the title, the purpose and the hypothesis. Accordingly, results are also focused better taking into account the title.
 
 
Point 4: Thus, the results are presented comparing perceived loneliness and social competence only in children with CP and DCD; such results for the rest of the participants are not presented. Moreover, this section does not include the results of the Pearson correlation analysis, which is listed in the subsection 2.5. In our opinion, it is need the correlation analysis between the studied variables (n = 165) that is necessary to achieve the purpose of the research: “to study the association between motor performance and perceived loneliness and social competence in 11-year-old children born very preterm”. 
 
Answer 4: Thank you for these important comments. We have now presented the results of typically developed children in Table 3. Accordingly, the results of the Pearsons correlation analysis has been added to the text:
The percentiles for the total scores of the Movement ABC-2 correlated negatively with perceived social loneliness (r=-0.2, p=0.049), when all the study children were included. The correlation did not remain statistically significant when the groups (CP, DCD, typical motor development) were analyzed separately.
 
 
Point 5: Accordingly, it is not clear on what basis the authors write in the Discussion that “in contrast to our hypothesis, motor performance did not correlate with perceived loneliness or social competence in children with typical motor development at 11 years of age.” Unfortunately, the article does not contain the results of the correlation analysis, as well as the results of the comparison between boys and girls, which are discussed below. 
 
Answer 5: This is an important observation. We apologize that essential information was missing from the manuscript. The correlation analysis has been added, please see the previous answer. The results of the comparison between boys and girls can be found at the pg 7:
In post-hoc analysis, boys born very preterm were more likely to report less empathy (boys mean 5.7 vs girls mean 6.3, p<0.001) and more disruptiveness (boys mean 3.9 vs girls mean 3.5, p=0.002) compared to girls born very preterm.
 
Point 6: In conclusion, I can say that the authors have good empirical data, but this version of the article does not provide a clear analysis of these data and a description of the results in accordance with the purpose and hypothesis of the study.  
 
Answer 6: We agree that there was discrepancy between the purpose and the results of the study. We have now clarified the objectives and hypothesis that are now congruent with the results. 
 
Reviewer 2 Report
Thank you for the opportunity to review this paper.
This study explores the relationship between perceived loneliness and social competence and motor impairment in 11 year old children born very preterm from a Finnish cohort, finding no significant correlation. Although no correlation was found, this paper asks an important question and provide new knowledge on outcomes of children born very preterm.
I have provided some comments below.
Introduction:
Overall clearly written and easy to read.
Methods:
pg. 2 lines 88-91: You’ve accidentally doubled up some text (just a quick edit needed).
pg.2 lines 95-99: Perhaps consider putting this information under a different heading, as doesn’t actually describe the participants, but the outcome measures assessed at 11.
Figure 1: As a reader I would be very interested to know the number of children who were VP vs VLBW in this cohort. Could this be included in this flow diagram?
Consider listing the reasons for withdrawal during follow-up.
pg 4 section 2.2: In the discussion (pg. 8, lines 272-77 approx) you provide some helpful information around why you chose your cut-off of <5th percentile, and reference the Blank 2019 DCD paper. Consider moving this information into this methods section instead.
pg. 4 sections 2.3 and 2.4: is there any additional information you could include here on how to interpret these measures? for example, minimal clinical important differences, or cut-off scores for concern, mean scores in the general population etc.
Results:
pg 7 lines 207-210: Comparison of MASCS scores between boys and girls wasn’t mentioned in the methods section, so there needs to be a discussion somewhere in the paper about why this analysis was done (and if post-hoc, clearly stated). If it was planned, it should be added to the methods.
pg 7 lines 214-219: The differences between children with DCD and CP in social and emotional loneliness is 0.1 and 0.2 on the PNDL. Does this actually constitute a difference? Particularly with such small numbers of participants. While I understand you’re using descriptive statistics here because of the small numbers, please be careful with the strength of language used. As per my comment in the methods- it would be really helpful to have some interpretation of these numbers (e.g. cut-off scores etc.).
Please consider these same points for the MASCS scores DCD vs CP.
Table 3: I realise this is in your methods, but please consider adding to the table a key to indicate if higher scores=better or worse outcomes for each variable. This will make is quicker and easier for readers to interpret the table.
Results in general:
Please consider adding the PNDL and MASCS scores for children born VP without DCD or CP somewhere to the results (perhaps table 3?). This will provide the reader with more information about loneliness and social competence for children born VP (even though the correlations were not statistically significant, this is still important information).
Discussion:
lines 228-231: Your first sentence of your discussion uses quite strong/definite language, but is based on descriptive statistics from a small number of children (see comment about lines 214-19). Please consider re-wording this sentence.
Please also consider leading the discussion with the results of your main aim (association btw motor outcomes and loneliness/social competence), as this is your primary result (and still important, even if your results weren’t significant).
pg 8 lines 270-287: Your limitations paragraph provides good description of the limitations of small numbers/sample sizes.
In interpreting your results, did you also consider the influence of the changing inclusion criteria (from VLBW to VP), as rates of poor outcomes across many developmental domains increase with lower gestational age (and many of your VLBW cohort would be considered moderate to late preterm, and thus less likely to have poor outcomes compared with children VP).
Finally, as mentioned in a few of my comments, I would really like some more information on how to interpret the PNDL and MASCS scores, or if this information is not available, some addition to the discussion on how this effects interpretation of your results.
Author Response
Dear Reviewer,
 
Thank you for improving the manuscript with your thorough comments. We have now answered the comments below. Accordingly, we have revised the manuscript carefully. 
 
 
Reviewer’s Comments to Author 
 
 
Point 1: This study explores the relationship between perceived loneliness and social competence and motor impairment in 11 year old children born very preterm from a Finnish cohort, finding no significant correlation. Although no correlation was found, this paper asks an important question and provide new knowledge on outcomes of children born very preterm.  
 
Answer 1: Thank you for your constructive comments. We have made changes to the manuscript to meet your suggestions to further improve the manuscript. All additions to the text are marked in red. 
 
Specific comments 
 
Methods:
Point 2: pg. 2 lines 88-91: You’ve accidentally doubled up some text (just a quick edit needed).
  
Answer 2: We apologize all the technical shortcomings that were related to compatible problems with apple computer and the template. All pictures and tables are available also as separate formats. We have revised the manuscript with pc-computer so there should be no more technical problems.  
 
 
Point 3: pg 2 lines 95-99: Perhaps consider putting this information under a different heading, as doesn’t actually describe the participants, but the outcome measures assessed at 11. 
 
Answer 3: Thank you for this comment. We now clarified the sentence of inclusion criteria:
In addition, the participants in the present study had to have the following evaluations at 11 years of age 1) Movement Assessment Battery for Children – Second Edition (Movement ABC-2) [36] for motor outcome, 2) Peer Network and Dyadic Loneliness scale (PNDL) [28,37] for perceived loneliness, and 3) Multisource Assessment of Children’s Social Competence Scale (MASCS) [38,39] for perceived social competence.
Point 4: Figure 1: As a reader I would be very interested to know the number of children who were VP vs VLBW in this cohort. Could this be included in this flow diagram. Consider listing the reasons for withdrawal during follow-up. 
 
Answer 4: We appreciate these observations. The number of children with VP and VLBW are now included to the text:
There were 143 children born <32 weeks of gestation and 154 children born ≤1500 grams.
Unfortunately, we have no exact data for the reasons for withdrawal of participants during follow-up, as according to ethical approval, the participants are allowed to withdraw at any time without giving the reason. 
 
Point 5: pg 4 section 2.2: In the discussion (pg. 8, lines 272-77 approx) you provide some helpful information around why you chose your cut-off of <5th percentile, and reference the Blank 2019 DCD paper. Consider moving this information into this methods section instead. 
 
Answer 5: Thank you for this comment. The information about cut off <5th percentile has been moved to methods section and the reference has been added. 
 
Point 6: pg. 4 sections 2.3 and 2.4: is there any additional information you could include here on how to interpret these measures? for example, minimal clinical important differences, or cut-off scores for concern, mean scores in the general population etc. 
 
Answer 6: Thank you for this important question. Cut-off and mean scores of the PNDL are defined separately for girls and boys. The mean scores in the general population have been defined at 2009. Later studies have been shown that loneliness has increased after that. Cut-off and mean scores of the MASCS are defined separately for girls, boys, children with learning disabilities and without learning disabilities. There are no norms for children born very preterm, neither recarding the PNDL nor MASCS. These are the reasons we could not show exact cutt-off scores and mean scores in this paper. Differences between children with CP and DCD were not statistically analyzed, because of the small number of cases. That is the reason we chose to show only descriptive statistics. We have now added that information to the discussion:
There are no norms or cut-offs for children born very preterm regarding the results of the PNDL and the MASCS.
Existing scores can be found in referred publications:
PNDL: Junttila, N. Vauras, M: Loneliness among School-Aged Children and Their Parents. Scandinavian Journal of Psychology 2009.
MASCS: Junttila, N.; Voeten, M.; Kaukiainen, A.; Vauras, M. Multisource Assessment of Children’s Social Competence. Educational and Psychological Measurement 2006
 
 
 Results:
Point 7: pg 7 lines 207-210: Comparison of MASCS scores between boys and girls wasn’t mentioned in the methods section, so there needs to be a discussion somewhere in the paper about why this analysis was done (and if post-hoc, clearly stated). If it was planned, it should be added to the methods.
Answer 7: Thank you for this good comment. We have clarified this to results section:
In post-hoc analysis, boys born very preterm were more likely to report less empathy (boys mean 5.7 vs girls mean 6.3, p<0.001) and more disruptiveness (boys mean 3.9 vs girls mean 3.5, p=0.002) compared to girls born very preterm.
Point 8: pg 7 lines 214-219: The differences between children with DCD and CP in social and emotional loneliness is 0.1 and 0.2 on the PNDL. Does this actually constitute a difference? Particularly with such small numbers of participants. While I understand you’re using descriptive statistics here because of the small numbers, please be careful with the strength of language used. As per my comment in the methods- it would be really helpful to have some interpretation of these numbers (e.g. cut-off scores etc.). Please consider these same points for the MASCS scores DCD vs CP.
Answer 8: Thank you for this important question. We have discussed the clinical relevance:
Children with DCD reported less perceived loneliness, but more problems with social competence compared to children with CP or typical motor development. The absolute differences in PNDL and MASCS results between the groups were minor. Whether these differences have clinical importance is not definite.
 
Point 9: Table 3: I realise this is in your methods, but please consider adding to the table a key to indicate if higher scores=better or worse outcomes for each variable. This will make is quicker and easier for readers to interpret the table.
 Answer 9: Thank you for this good point. The addition you suggested has been made in Table 3.
Results in general:
Point 10: Please consider adding the PNDL and MASCS scores for children born VP without DCD or CP somewhere to the results (perhaps table 3?). This will provide the reader with more information about loneliness and social competence for children born VP (even though the correlations were not statistically significant, this is still important information).
Answer 10: We appreciate these observations. We have now presented the results of typically developed children in Table 3.
 
Discussion:
Point 11: lines 228-231: Your first sentence of your discussion uses quite strong/definite language, but is based on descriptive statistics from a small number of children (see comment about lines 214-19). Please consider re-wording this sentence.
Answer 11: We have re-worded the sentences:
Children with DCD reported less perceived loneliness, but more problems with social competence compared to children with CP or with typical motor development. The absolute differences in PNDL and MASCS results between the groups were minor. Whether these differences have clinical importance is not definite.
Point 12: Please also consider leading the discussion with the results of your main aim (association btw motor outcomes and loneliness/social competence), as this is your primary result (and still important, even if your results weren’t significant).
Answer 12: We have edited the discussion according your relevant suggestion.
 
Point 13: pg 8 lines 270-287: Your limitations paragraph provides good description of the limitations of small numbers/sample sizes
Answer 13:. We appreciate this comment.
Point 14: In interpreting your results, did you also consider the influence of the changing inclusion criteria (from VLBW to VP), as rates of poor outcomes across many developmental domains increase with lower gestational age (and many of your VLBW cohort would be considered moderate to late preterm, and thus less likely to have poor outcomes compared with children VP).
Answer 14: Thank you for this comment. The limited number of children with motor impairment did not permit adjusted analyses but we have previously shown that very preterm children born SGA have similar neurodevelopmental outcomes compared with non-SGA very preterm children.
References:
-Lind A, Parkkola R, Lehtonen L, et al. Associations between regional brain volumes at term-equivalent age and development at 2 years of age in preterm children. Pediatr Radiol 2011;
-Maunu J, Lehtonen L, Lapinleimu H, et al. Ventricular dilatation in relation to outcome at 2 years of age in very preterm infants: a prospective Finnish cohort study. Dev Med Child Neurol 2011
-Leppanen M, Lapinleimu H, Lind A, et al. Antenatal and postnatal growth and 5-year cognitive outcome in very preterm infants. Pediatrics 2014
Point 15: Finally, as mentioned in a few of my comments, I would really like some more information on how to interpret the PNDL and MASCS scores, or if this information is not available, some addition to the discussion on how this effects interpretation of your results.
Answer 15: Thank you for these observations. The same answer as above:
Cut-off and mean scores of the PNDL are defined separately for girls and boys. The mean scores in the general population have been defined at 2009. Later studies have been shown that loneliness has increased after that. Cut-off and mean scores of the MASCS are defined separately for girls, boys, children with learning disabilities and without learning disabilities. There are no norms for children born very preterm, neither recarding the PNDL nor MASCS. These are the reasons we could not show exact cutt-off scores and mean scores in this paper. We have now added that information to the discussion:
There are no norms or cut-offs for children born very preterm regarding the results of the PNDL and the MASCS.
Existing scores can be found in referred publications:
PNDL: Junttila, N. Vauras, M: Loneliness among School-Aged Children and Their Parents. Scandinavian Journal of Psychology 2009.
MASCS: Junttila, N.; Voeten, M.; Kaukiainen, A.; Vauras, M. Multisource Assessment of Children’s Social Competence. Educational and Psychological Measurement 2006
 
 
 
  
 
 
 

Round 2
Reviewer 1 Report
Dear authors,
I am grateful that you took into account most of my comments and recommendations and significantly improved the manuscript.
I agree with all corrections and additions in the new version of the manuscript.
However, I have a few more small suggestions to improve the manuscript:
- I recommend adding a statistical analysis to the data presented in Table 3
- I still recommend adding tables with the results of correlation analysis to the article, even if no statistically significant correlations were obtained.
Author Response
Dear Reviewer,
Thank you for improving the manuscript with your further suggestions. We have now answered the comments and revised the manuscript accordingly. All additions are marked in red.
 
Reviewers' Comments to Author 
 
Point 1: I recommend adding a statistical analysis to the data presented in Table 3 
Answer 1: We have now added the statistical analysis in Table 4 (former Table 3) and methods.
 
 
Point 2: I still recommend adding tables with the results of correlation analysis to the article, even if no statistically significant correlations were obtained.
Answer: Thank you for this comment. We have now added the table with the results of correlation analysis (Table 3).
Best regards,
Minttu Helin